# Enhanced Nuclei Segmentation and Classification via Category Descriptors in the SAM Model

**DOI:** 10.3390/bioengineering11030294

**Published:** 2024-03-21

**Authors:** Miguel Luna, Philip Chikontwe, Sang Hyun Park

**Affiliations:** 1Department of Robotics and Mechatronics Engineering, Daegu Gyeongbuk Institute of Science and Technology (DGIST), Daegu 42988, Republic of Korea; miguel@dgist.ac.kr (M.L.); philipchicco@dgist.ac.kr (P.C.); 2AI Graduate School, Daegu Gyeongbuk Institute of Science and Technology (DGIST), Daegu 42988, Republic of Korea

**Keywords:** nuclei segmentation, nuclei classification, prompt guided segmentation, domain alignment, long-tailed distribution

## Abstract

Segmenting and classifying nuclei in H&E histopathology images is often limited by the long-tailed distribution of nuclei types. However, the strong generalization ability of image segmentation foundation models like the Segment Anything Model (SAM) can help improve the detection quality of rare types of nuclei. In this work, we introduce category descriptors to perform nuclei segmentation and classification by prompting the SAM model. We close the domain gap between histopathology and natural scene images by aligning features in low-level space while preserving the high-level representations of SAM. We performed extensive experiments on the Lizard dataset, validating the ability of our model to perform automatic nuclei segmentation and classification, especially for rare nuclei types, where achieved a significant detection improvement in the F1 score of up to 12%. Our model also maintains compatibility with manual point prompts for interactive refinement during inference without requiring any additional training.

## 1. Introduction

Performing analysis of the micro-environment in histopathology samples is a crucial step to understand the status and prognosis of cancer tumors [1,2,3,4]. The presence of eosinophils in tumor sites and the neutrophil-to-lymphocyte ratio have already been used as prognostic indicators in oncologic clinical practice [5,6]. Also, high numbers of tumor-infiltrating lymphocytes have been connected to the inhibition of tumor progression [7], and plasma cells are known to secrete high amounts of antibodies, protecting the host against toxins and pathogens [8]. However, automatic detection of some *rare* types of nuclei is challenging due to the long-tailed distribution of nuclei in tissue samples and the relatively small size of available datasets. Deep learning methods [9,10,11,12,13,14,15,16,17] have demonstrated great ability to automatically extract meaningfully features from data, but their performances have been limited by the relatively small size of datasets [18,19,20,21]. In contrast, foundational models [22,23,24,25] have shown better generalization by training on very large datasets. Thus, we identify the strong representation of foundational models as a way to overcome the issues derived from the long-tailed distribution of histopathology images.

Recently, the release of the Segment Anything Model (SAM) [26] has opened the possibility to use foundation models for image segmentation. The model has been trained on the SA-1B dataset containing 11 M natural scene images with 1 B masks. The large amount of labeled data has allowed the model to learn a strong representation to detect complex patterns in order to segment a wide variety of objects. In order to make predictions, the SAM model uses point, bounding box, and mask prompts to return valid segmentation masks in an interactive way. Therefore, several prompts can be combined to let the model identify foreground segments and reject others. In this way, we hypothesize that given the right combination of prompts, the model could segment nuclei foreground, background tissue, or even nuclei boundary pixels (a technique commonly used for nuclei segmentation [27,28,29,30]). The advantage of using prompts over task-specific tuning of the output layers is that the model’s learned representations are preserved, leading to better generalization and preventing overfitting [31].

In this work, we introduce a category prompt encoder to learn category descriptors for each type of nuclei, background tissue and nuclei boundaries. In Figure 1, we show that category descriptors applied to the SAM mask decoder generate different segmentation masks based on the prompts used. We also show that the existing domain gap between histopathology and natural scene images limits the performance of the vanilla SAM model. Therefore, we introduce a domain alignment module to close the gap leading to better quality segmentation outputs. Instead of adding adapter layers to the transformer blocks of the model [32,33], we only adapt features in low-level space to preserve the strong representation of the model. Our experimental results show the significantly improved detection ability of our model, especially on *rare* types of nuclei. Moreover, our model also maintains compatibility with point prompts, allowing interactive refinement at inference time even though no point prompts were used during training; thus, this demonstrates that our domain alignment module effectively adapts the SAM model to histopathology images while preserving the model’s internal representation. We summarize our main contributions as follows:We introduce category descriptors to perform automatic nuclei segmentation and classification via prompting the SAM model.We align the low-level features of histopathology images with the distribution of natural scenes features to exploit the high-level representation of the SAM model for accurate nuclei segmentation and classification.We also show that the inherent ability of the SAM model is still preserved after domain alignment and can use manual point prompts (not used during training) on histopathology images for further interactive refinement during inference.

In the following sections, we introduce the relevant literature in Section 2, describe our methodology in Section 3, introduce the datasets and experimental settings in Section 4, and provide the experimental results and ablations in Section 5.

## 2. Related Works

### 2.1. Nuclei Segmentation

Nuclei segmentation was initially performed by first detecting foreground pixels and later applying post-processing algorithms to separate individual cells [34,35,36,37]. Subsequent works included boundary detection to allow models learn patterns to separate touching nuclei using a three-class detection task [27,29,30]. In contrast, other works used a regression task to determine the boundaries between nuclei. Naylor et al. [20] used the distance map, while Graham et al. [38] encoded nuclear instances into vertical and horizontal distances in order to determine their centers and boundaries. However, He et al. [12] showed that adding stronger emphasis on boundary classification led to higher nuclei segmentation performance using the three-class detection task. In our work, we use category descriptors to prompt the SAM model in order to determine nuclei foreground per class, boundary pixels, and background.

### 2.2. Segment Anything Model (SAM)

With the release of the SAM model [26], several works have focused on leveraging the strong representation of the foundation model to perform tasks in unseen domain including medical images. Zhang et al. [39] used the predictions produced by the SAM model to augment medical images to train a task-specific segmentation model. Ma et al. [40] used bounding box prompts to perform segmentation in medical images and fine-tuned both the image encoder and the mask decoder while keeping the prompt encoder fixed. Mazurowski et al. [41] studied the zero-shot accuracy of the SAM model in medical image segmentation. They found that SAM performs better on well-circumscribed objects and the benefit of using additional point prompts is limited. Furthermore, Huang et al. [42] found that combining point and bounding box prompts performs better on medical images. Also, fine-tuning the mask decoder shows improvements but performance on small or rare objects decreases.

Due to domain gaps between the SA-1B dataset and the target datasets, several works have explored ways to fine-tune the model to the increase performance of the target task. Most works have opted to include adapter modules in the image encoder and mask decoder. Xiong et al. [43] added convolutional adapters between transformer layers with a custom multi-scale decoder for segmentation output. Chen et al. [44] also used adapter modules between transformer layers in the image encoder where conditional prompts were directly applied for each specific task, with the mask decoder also fine-tuned. Pu et al. [31] added adapter modules to each transformer module in the image encoder and fully trained a custom decoder. Wu et al. [45] included adapter modules to each transformer layer in both the image encoder and mask decoder modules. However, adding trainable parameters to modify high-level features incurs the risk of over-fitting on the training dataset, reducing the inherent ability of the foundation model. In contrast, we perform domain alignment in low-level feature space, preserving the strong representation learned by the SAM model on the SA-1B dataset.

## 3. Method

The SAM model has acquired strong high-level representations for segmenting a wide diversity of objects by training with over 1 billion masks. However, it is not possible to apply the vanilla SAM model to histopathology images for automatic nuclei segmentation and classification. The SAM model relies on manual prompts to interactively perform segmentation while the grid of point prompts used in the “everything mode” suffers from ambiguous boundaries commonly encountered in histopathology images. In addition, there is no built-in capability to perform classification. Therefore, we devise a prompting scheme with category descriptors to segment and classify nuclei while preserving the high-level representation of the SAM model. We exploit the ability of the model to run multiple prompts at low cost to extract several prediction masks. In detail, we predict masks for each type of nuclei as well as background tissue and nuclei boundaries to help separate individual instances. By combining these predicted masks, we can accurately segment nuclei of different types. Although the vanilla SAM model can perform nuclei segmentation and classification with category descriptors, the domain gap between histopathology and natural scenes images limits the performance of the model. Thus, we introduce a domain alignment module to project the low-level features of the histopathology images into a space closer to natural scene features where the high-level representation of the model could be better exploited. In Figure 1, we present the architecture details of our model and provide visual examples on how the predicted masks quality is improved by our category descriptors and domain alignment module.

**Figure 1 bioengineering-11-00294-f001:**
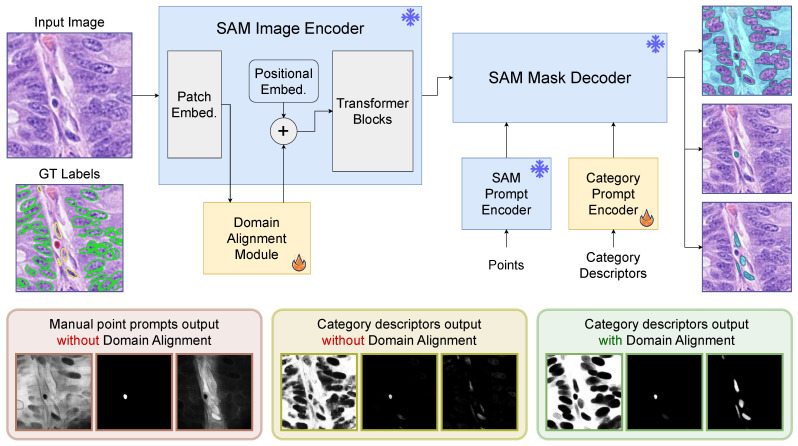
Our proposed category descriptors are an effective way to perform automatic nuclei segmentation and classification. Point prompts can be used for interactive manual refinement during inference, but they are not used at training time. Domain alignment in low-level feature space is used to bridge the gap between histopathology and natural scene images while preserving the high-level representation of the model. Category descriptors alone demonstrate superior segmentation ability over manual prompts while domain alignment enhances the segmentation quality.

### 3.1. Category Descriptors

The combination of multiple prompts allow the SAM model to predict complex segmentation masks. Prompts are transformed into tokens and stacked together to let the mask decoder recover the mask of a target object. In a similar way, we define a set of learnable category descriptors in the form of a stack of tokens for each nuclei type, background and boundary pixels. The number of tokens per class is studied in Section 5.1. In Figure 1, we demonstrate that the vanilla SAM model is able to perform segmentation of some types of nuclei via manual prompts, but the lack of clear boundaries severely affects the output masks. In contrast, using our learned category descriptors allow the mask decoder to recover fairly accurate masks, but due to the domain gap, predictions are still affected by noise and the confidence of some masks is very low. Mitigating the domain gap lets the model predict very clear and confident masks across nuclei types using our category descriptors.

### 3.2. Domain Alignment

Fine-tuning is a common way to transfer the representation learned on large datasets to smaller ones on relatively similar tasks. However, updating the parameters of a foundational model might hurt the high-level learned representation leading to over-fitting. Instead, we propose to perform domain alignment in low-level feature space. This allows the model to shift the external domain image features to a space where the high-level patterns are more effective for the target task. As depicted in Figure 1, we insert a domain alignment module between the patch embedding layer and the transformer blocks. Specifically, the patch embedding layer maps the image to a larger dimensional space for feature extraction. Then, we use a sequence of lightweight residual layers to project the image features to an optimal distribution in low-level space. Each residual layer shifts the low-level features from the input histopathology image to a distribution closer to natural scene images that is expected by the transformer blocks. In this way, we preserve the high-level representation of the SAM model to perform nuclei segmentation and classification via category descriptors. Another advantage is that point prompts can be also applied through the SAM prompt encoder without any additional training (an ablation is conducted in Section 5.2). Finally, in order to minimize the size of our domain alignment module, we use lightweight inverted residual layers following the MobileNetV2 implementation [46] to reduce the module memory footprint while retaining a strong performance.

### 3.3. Training Objective

Each set of category descriptors are learned to activate an independent type of nuclei and could be optimized separately from other descriptors. However, due to morphological similarities between different types of nuclei, it is important to emphasize negative gradients depending on other classes predictions. Therefore, we consider two fundamental aspects to ensure *rare* classes have higher activation than *frequent* ones in ambiguous cases. First, we employ a federated style loss [47] to ensure that negative gradients are only applied to classes that appear in the image reducing their impact on *rare* classes. Second, we adopt a multi-class hinge loss strategy to focus on hard samples that have high activation among several category descriptors. This allow us to reduce negative unnecessary gradients from easier samples. In detail, we employ a separate binary cross entropy loss for each category that is only applied to cell types present in the image. However, the loss has a margin γ that compares predictions across all categories and only back-propagates gradients for ambiguous samples ignoring confident pixels. In our experiments, we set γ=0.2 as the minimum prediction probability gap between the target category and other categories.

## 4. Experiments

### 4.1. Dataset

We run our experiments on the Lizard dataset [48]. The dataset contains images from a wide variety of colon tissue samples including normal, inflammatory, dysplastic, and cancerous conditions. The images were extracted from colon H&E images at 20× objective magnification (∼0.5 µm/pixel). The publicly available version of the dataset is divided in subsets obtained from five different sources: CoNSeP [38], CRAG [49], DigestPath [50], GlaS [51], and PanNuke [52]. The dataset was extensively annotated with 495,179 individual nuclei masks separated across six different types of cells, i.e., neutrophil, eosinophil, plasma, connective, lymphocyte, and epithelial. However, 92.4% of the instances are epithelial, lymphocyte, and connective cells (*frequent* categories) while only 5.9% are plasma cells and the remaining 1.7% cells are neutrophils and eosinophils (*rare* categories). Thus, due to the long-tailed distribution of cell types, we pay special attention to the accurate detection of *rare* categories in our experiments.

### 4.2. Experimental Setup

The majority of labeled nuclei come from the DigestPath and CRAG subsets, whereas the fewest labeled nuclei come from the CoNSeP subset. Thus, in our experiments we selected the CoNSeP subset as our validation set and performed four-fold cross-validation with the remaining four subsets for testing purposes (CRAG, DigestPath, GlaS, PanNuke). The results on the CoNSeP validation set were computed by averaging the scores of all four models trained during the four-fold cross-validation process and are reported in Table 1 and Table 2. Similarly, the test scores were computed for four different test subsets according to the four-fold cross-validation procedure, and we report the average scores in Table 3 and Table 4. Using cross-validation allowed us to test the resilience of the models to changes in the training and testing domains.

### 4.3. Evaluation Metrics

Due to the long-tailed distribution of the dataset, we focus on the detection accuracy across different types of nuclei. To this end, we employ the object-based F1 score, where true positives (TP) are defined by assigning every prediction to its closest ground truth label, and only one prediction is allowed for each ground truth mask. Predictions and labels without matching pair are treated as false positives (FP) and false negatives (FN), respectively. Prediction and label pairs are assigned according to the highest intersection over union score (IoU).
(1)$$\begin{array}{c} F1\;score=\frac{TP}{TP\;+\;\frac{1}{2}\left(FP+FN\right)} \end{array}$$
In addition, we evaluate the segmentation and classification combined performance using the mean average precision metric (mAP), commonly used for instance segmentation tasks:(2)$$\begin{array}{c} mAP=\frac{1}{N}\sum_{k=1}^{N}AP_{k}, \end{array}$$
where *k* is the nuclei category and AP (Average Precision) is the area under the precision–recall curve. Specifically, we use the MS COCO evaluation algorithm (https://github.com/cocodataset/cocoapi (accessed on 21 December 2023)) with 101 interpolation points and 10 IoU thresholds on the precision–recall curve to compute the average precision (AP) for a more fine-grained evaluation.

### 4.4. Comparison Methods

We compare our method with a state-of-the-art nuclei segmentation method and a widely used instance segmentation method. CDNet [12] was originally proposed to perform nuclei segmentation without accounting for nuclei type classification. Thus, we extended the network with a classification branch following structure of the existing “mask branch”. However, to account for the long-tailed distribution of the Lizard dataset, we implemented two variants with different classification objectives. The Seesaw [53] and ECM [54] losses were developed to mitigate the negative effects of *frequent* categories over *rare* ones, making them suitable for our experiments.

Mask R-CNN [9] is a powerful segmentation method that has been widely applied to multiple instance segmentation tasks. We used the implementation provided by the Open MMLab Detection Toolbox [55], where the model has been highly optimized and pre-trained on the MS-COCO [21] dataset, providing a strong initial learned representation. We also run experiments using the Mask R-CNN model with the Seesaw loss [53] to account for the long-tailed distribution of the Lizard dataset.

### 4.5. Implementation Details

We used the official implementation of the SAM model (https://github.com/facebookresearch/segment-anything (accessed on 21 December 2023)) and ran our experiments using the ViT-B version of the model, i.e., the smallest released model with 91 M parameters. Our learnable category descriptors were defined with the same size as other prompt tokens with 256 channels. The inverted residual blocks used in the domain alignment module were reduced to 96 channels, where a 3×3 channel-wise convolution was applied. As shown in Figure 1, the original SAM image encoder, prompt encoder and mask decoder were not updated during training. We trained our model for 20 epochs using an Adam optimizer with learning rate of $10^{-3}$ with linear warm-up and step decay to $10^{-4}$ for the last 2 epochs.

We trained our model using randomly extracted image crops of size 256×256 pixels. We used repeat factor sampling (RFS) [56] to balance the rate at which *rare* and *frequent* categories were observed by the model. We also applied random flip, rotation, and color jittering augmentations to extend the variety of distributions seen by the model. For inference, we used a sliding window approach with a step size of 128 pixels to extract image crops of size 256×256 pixels and made separate predictions for each crop. We reconstructed the entire slide by adding the predicted instances at the center of each crop to a full-size prediction map without duplicating nuclei from neighboring crops. All metrics were computed using the full-size predictions and labels.

## 5. Results

Our experimental results demonstrate that our approach shows superior performance for segmenting and classifying nuclei, especially *rare* categories. In Figure 2, we show tissue samples containing neutrophils and eosinophils, where most models had limited detection results. These qualitative results show that Mask R-CNN models suffer from lower detection ability while CDNet models tend to assign classes incorrectly. In contrast, our category descriptors leverage the high-level representation of the SAM model to correctly segment and classify both *rare* and *frequent* nuclei.

In our quantitative results, we report the average performance of the four models obtained from our four cross-validation experiments. The results on the common validation set for all experiments (CoNSeP subset) are shown in Table 1 and Table 2. Our method shows significantly higher detection performance (F1 score) on *rare* categories while the gains on *frequent* categories are less pronounced. In fact, the object-based F1 score does not consider pixel level matching between prediction and label masks, therefore highlighting the ability of our model to assign the correct class to detected instances across all cell types. On the mAP evaluation, however, other methods might achieve higher scores on *frequent* categories due to higher pixel-level accuracy, but our approach consistently achieves higher segmentation and classification accuracy on *rare* categories.

The results on the test set add another variable to the evaluation of the models performance. Evaluating in considerable larger subsets than the CoNSeP subset, the generalization ability of the models plays a bigger role due to larger domain gaps between subsets. As shown in Table 3 and Table 4, the performance scores significantly reduced across all models. However, there is a consistent trend regarding the performance on *rare* and *frequent* categories. Our model significantly outperforms the segmentation and classification capabilities of other methods. In addition, due to the larger domain shift between subsets, the strong representation acquired by SAM enables better generalization over competing models even in the case of *frequent* classes.

**Table 1 bioengineering-11-00294-t001:** Validation F1 scores per object in the CoNSeP subset.

Method	F1_*mean*_	F1_*category*_					
		**Rare**			**Frequent**		
		**Neutrophil**	**Eosinophil**	**Plasma**	**Connective**	**Lymphocyte**	**Epithelial**
CDNet [12] + $L_{ecm}$ [54]	0.624	0.304	0.627	0.509	0.729	0.757	0.814
CDNet [12] + $L_{seesaw}$ [53]	0.671	0.443	0.690	0.574	0.725	0.768	0.824
Mask R-CNN [9]	0.665	0.382	0.646	0.564	0.781	0.788	0.827
MRCNN [9] + $L_{seesaw}$ [53]	0.668	0.382	0.676	0.556	0.78	0.782	0.829
Ours	0.733	0.540	0.797	0.645	0.785	0.789	0.844

**Table 2 bioengineering-11-00294-t002:** Validation mean average precision scores in the CoNSeP subset.

Method	mAP	mAP_50_	mAP_*category*_					
			**Rare**			**Frequent**		
			**Neutrophil**	**Eosinophil**	**Plasma**	**Connective**	**Lymphocyte**	**Epithelial**
CDNet [12] + $L_{ecm}$ [54]	0.225	0.423	0.064	0.171	0.19	0.236	0.385	0.303
CDNet [12] + $L_{seesaw}$ [53]	0.295	0.545	0.123	0.238	0.268	0.374	0.415	0.350
Mask R-CNN [9]	0.240	0.459	0.110	0.201	0.224	0.226	0.384	0.296
MRCNN [9] + $L_{seesaw}$ [53]	0.292	0.536	0.109	0.231	0.248	0.384	0.422	0.355
Ours	0.321	0.594	0.238	0.319	0.274	0.373	0.38	0.342

**Table 3 bioengineering-11-00294-t003:** Testing F1 scores per object applying 4-fold cross-validation on the CRAG, DigestPath, GlaS, and PanNuke subsets.

Method	F1_*mean*_	F1_*category*_					
		**Rare**			**Frequent**		
		**Neutrophil**	**Eosinophil**	**Plasma**	**Connective**	**Lymphocyte**	**Epithelial**
CDNet [12] + $L_{ecm}$ [54]	0.507	0.154	0.334	0.418	0.656	0.689	0.789
CDNet [12] + $L_{seesaw}$ [53]	0.565	0.236	0.460	0.465	0.699	0.710	0.819
Mask R-CNN [9]	0.533	0.244	0.394	0.482	0.619	0.722	0.739
MRCNN [9] + $L_{seesaw}$ [53]	0.521	0.219	0.371	0.478	0.616	0.714	0.730
Ours	0.639	0.371	0.590	0.565	0.735	0.731	0.839

**Table 4 bioengineering-11-00294-t004:** Testing mean average precision scores applying 4-fold cross-validation on the CRAG, DigestPath, GlaS, and PanNuke subsets.

Method	mAP	mAP_50_	AP_*category*_					
			**Rare**			**Frequent**		
			**Neutrophil**	**Eosinophil**	**Plasma**	**Connective**	**Lymphocyte**	**Epithelial**
CDNet [12] + $L_{ecm}$ [54]	0.162	0.309	0.021	0.038	0.128	0.223	0.320	0.246
CDNet [12] + $L_{seesaw}$ [53]	0.171	0.336	0.021	0.048	0.141	0.245	0.319	0.251
Mask R-CNN [9]	0.225	0.396	0.086	0.126	0.196	0.281	0.384	0.276
MRCNN [9]+$L_{seesaw}$ [53]	0.220	0.390	0.081	0.115	0.205	0.277	0.378	0.263
Ours	0.269	0.470	0.108	0.171	0.239	0.348	0.393	0.350

We evaluate the statistical significance of our four-fold cross validation results by applying the Wilcoxon signed-rank test. We verify that the underlying distribution of the difference of the paired samples between our model and comparison methods results is greater than a distribution symmetric about zero. Table 5 demonstrates that the differences are significant (p≤0.05) across all methods.

### 5.1. Ablation Studies

We run ablation studies on the number of category descriptors and residual layers used in our domain alignment module that are necessary to allow the SAM model segment and classify nuclei accurately. In our ablation experiments we use the GlaS and PanNuke subsets as training set and the CoNSeP subset for testing. In Table 6, we show the performance obtained at different number of residual alignment layers and category descriptors sizes. The case when there are zero residual alignment layers is equivalent to using our proposed category descriptors to prompt the original SAM model.

The results indicate that there is a positive relation between the number of category descriptors and the performance on the SAM model. However, without residual alignment layers, the detection quality is poor, and significantly increasing the number of category descriptors only leads to marginal gains. On the other hand, adding our domain alignment module results in a greater performance increase while reducing the number of required category descriptors per class.

As shown in Table 6, there is a tangential increase in performance beyond eight residual layers. Although using a larger number of parameters leads to additional performance gains, the GPU memory footprint also increases at a faster rate. Thus, we selected 12 residual alignment layers and 32 category descriptors for all our experiments as they show a reasonable trade-off between performance and memory requirement.

### 5.2. Manual Prompts

The vanilla SAM model allow additional interactions with the segmentation output through manual prompts. In our work, although point prompts were not used while training the domain alignment module (only prompts from category descriptors), in Table 7 we show that manual point prompts are still an effective way to make interactive corrections. Specifically, we tested the performance of our model by adding an increasing number of manual point prompts per nuclei type, i.e., a single point prompt per cell type is added at each step. For this process, we used the ground truth labels to select one instance per class that has the lowest confidence to add the next point prompt. The results show that *rare* categories achieve major improvements while *frequent* nuclei types only have marginal changes. Therefore, preserving compatibility with point prompts validates our approach to perform domain alignment in low-level feature space without affecting the strong high-level representation of the SAM model.

## 6. Limitations

Although leveraging the strong learned representation of SAM helps to mitigate the problem of classifying nuclei under a long-tailed distribution, further research is needed to decrease the computational demands of such a large foundational model. Considering that whole-slide histopathology images are usually very large, in the order of Gigapixels, smaller and memory efficient models will be required for practical use. In this sense, knowledge distillation techniques [57] are a viable option for smaller models to learn adequate representations from large teacher models. Future research on the topic is still required.

## 7. Generalizability

Our ablation results (Table 6) demonstrate that a reasonable number of residual alignment layers combined with category descriptors are sufficient to adapt SAM to alternative tasks such as nuclei segmentation and classification, while addressing the domain gap with medical images. We believe our technique can be extended to segment and classify other types of medical images by defining adequate prompts (category descriptors) and objective functions according to the target task.

## 8. Conclusions

In this work, we showed that the SAM model already has a powerful learned representation that enables it to perform segmentation on unrelated domains such as nuclei segmentation in H&E Histopathology images. Performing proper domain alignment in low-level feature space allowed the us to leverage the SAM model to accurately detect different types of nuclei. Moreover, learning separate category prompts showed to be an effective way to classify nuclei under a long-tailed distribution. Further statistical analysis confirmed the superiority of the results obtained by our model. In addition to performing automatic nuclei segmentation and classification, we also highlight that our model can refine predictions with the aid of manual prompts that significantly improve the quality of model outputs. Although we achieved great improvements on detection of neutrophil and eosinophil nuclei types (*rare* categories), performance improvement was less pronounced in plasma cells due to similar appearance with lymphocytes (a *frequent* type of nuclei). Further investigation on what other factors can be leveraged from foundation models to distinguish nuclei types with highly similar morphology is left for future work.

## Figures and Tables

**Figure 2 bioengineering-11-00294-f002:**
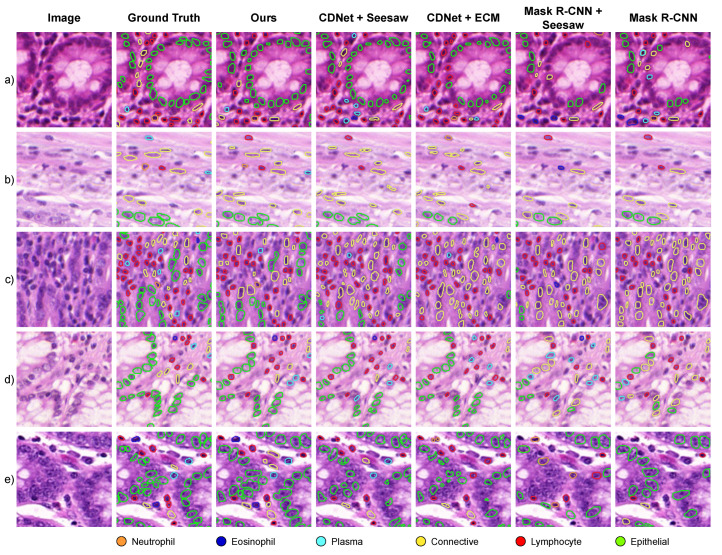
Segmentation and classification results across different tissue samples containing *rare* nuclei types. Neutrophils are shown in (**a**–**c**) rows, while Eosinophils in (**d**,**e**).

**Table 5 bioengineering-11-00294-t005:** Wilcoxon signed-rank test of the difference of the paired samples between our model and comparison methods.

Method	CDNet [12] + $L_{\mathrm{ecm}}$ [54]	CDNet [12] + $L_{\mathrm{seesaw}}$ [53]	Mask R-CNN [9]	MRCNN [9] + $L_{\mathrm{seesaw}}$ [53]
*p*-value	$8.6\times 10^{-23}$	$1.5\times 10^{-21}$	$8.8\times 10^{-16}$	$3.3\times 10^{-16}$

**Table 6 bioengineering-11-00294-t006:** Ablation results on the number of residual alignment layers and the size of the category descriptors. These experiments were run training on GlaS and PanNuke subsets and testing on the CoNSeP subset.

# Residual Layers	# Category Descriptors	F1	mAP
0	8	0.226	0.046
0	32	0.285	0.060
0	128	0.308	0.069
0	512	0.309	0.079
4	32	0.498	0.161
8	32	0.628	0.237
12	32	0.638	0.240
16	32	0.643	0.244
12	8	0.629	0.235
12	16	0.630	0.230
12	32	0.638	0.240
12	64	0.645	0.242
12	128	0.647	0.246

**Table 7 bioengineering-11-00294-t007:** Ablations on the use of manual point prompts for a model trained on GlaS and PanNuke subsets and tested on the CoNSeP subset.

# of Prompts	F1_*mean*_	F1_*category*_					
		**Rare**			**Frequent**		
		**Neutrophil**	**Eosinophil**	**Plasma**	**Connective**	**Lymphocyte**	**Epithelial**
0	0.638	0.621	0.565	0.412	0.694	0.751	0.783
1	0.699	0.758	0.667	0.488	0.717	0.779	0.788
2	0.725	0.806	0.731	0.504	0.728	0.787	0.792
4	0.747	0.866	0.755	0.528	0.738	0.799	0.794
8	0.754	0.866	0.764	0.542	0.749	0.809	0.796
16	0.762	0.879	0.771	0.556	0.756	0.814	0.796

## Data Availability

We used publicly available data.

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
