# Peer review of "Enhanced Nuclei Segmentation and Classification via Category Descriptors in the SAM Model"

_bioengineering, 2024, doi:10.3390/bioengineering11030294_

Round 1

Reviewer 1 Report

Comments and Suggestions for Authors

Abstract should be updated to include some numerical results.

Add one paragraph to describe the structure of the paper by the end of section 1.

In section 2, the authors are advised to enrich the discussion of the important literature works in deep learning (“Mohamed, M. (2023) “Empowering deep learning based organizational decision making: A Survey”, Sustainable Machine Intelligence Journal, 3. doi: 10.61185/SMIJ.2023.33105.”, “Mohamed, M. (2023) “Agricultural Sustainability in the Age of Deep Learning: Current Trends, Challenges, and Future Trajectories”, Sustainable Machine Intelligence Journal, 4, p. (2):1−20. doi: 10.61185/SMIJ.2023.44102.”) through discussing them in introduction/literature or include them within experimental comparisons.

in section 4.3, evaluation metrics miss math expressions.

it's highly recommended to use cross-validation in you experiments.

Experimental comparisons miss statistical analysis

Comments on the Quality of English Language

The authors should strive for greater clarity and precision in their use of the English language.

Reviewer 2 Report

Comments and Suggestions for Authors

The authors have enhanced the SAM (Segment Everything Model), a recently published model, by incorporating predicted masks to facilitate histopathology image segmentation. While the innovation in the method appears somewhat limited, its adaptability to various image types is acknowledged. I recommend including the original SAM as one compression method to augment the paper, considering that SAM may have been initially trained using the Lizard dataset. This addition would provide valuable context and insight into the model's evolution, enhancing the study's comprehensiveness.

Round 2

Reviewer 1 Report

Comments and Suggestions for Authors

- The limitations should be pointed out before conclusion

- The authors are advised to address generalizability of the proposed method

Comments on the Quality of English Language

good
